# Recent Progress in Metal–Organic Framework (MOF) Based Luminescent Chemodosimeters

**DOI:** 10.3390/nano9070974

**Published:** 2019-07-03

**Authors:** Yuanqiang Hao, Shu Chen, Yanli Zhou, Yintang Zhang, Maotian Xu

**Affiliations:** 1Henan Key Laboratory of Biomolecular Recognition and Sensing, College of Chemistry and Chemical Engineering, Shangqiu Normal University, Shangqiu 476000, China; 2Key Laboratory of Theoretical Organic Chemistry and Function Molecule of Ministry of Education, School of Chemistry and Chemical Engineering, Hunan University of Science and Technology, Xiangtan 411201, China; 3College of Chemistry and Molecular Engineering, Zhengzhou University, Zhengzhou 450001, China

**Keywords:** metal–organic frameworks, luminescent sensor, chemodosimeter, review

## Abstract

Metal–organic frameworks (MOFs), as a class of crystalline hybrid architectures, consist of metal ions and organic ligands and have displayed great potential in luminescent sensing applications due to their tunable structures and unique photophysical properties. Until now, many studies have been reported on the development of MOF-based luminescent sensors, which can be classified into two major categories: MOF chemosensors based on reversible host–guest interactions and MOF chemodosimeters based on the irreversible reactions between targets with a probe. In this review, we summarize the recently developed luminescent MOF-based chemodosimeters for various analytes, including H_2_S, HClO, biothiols, fluoride ions, redox-active biomolecules, Hg^2+^, and CN^−^. In addition, some remaining challenges and future perspectives in this area are also discussed.

## 1. Introduction

Metal–organic frameworks (MOFs), constructed by metal ions (or clusters) with organic ligands, are a subclass of coordination polymers with highly crystalline structures [1,2]. In the past two decades, enormous progress has been made in the synthesis and application of MOFs [3,4,5]. Because of their remarkable structural features, such as high porosity and large surface area, as well as their exceptional physicochemical properties, MOFs possess versatile applications in gas storage and separation [6,7,8,9,10,11], heterogeneous catalysis [12,13,14,15], drug delivery and chemotherapy [16,17], and sensing [18,19]. As new types of promising functional materials, luminescent MOFs especially have great potential as alternative phosphors in lighting devices and luminescent sensors [20,21,22,23].

Due to the versatile building blocks of MOFs (inorganic ions and organic ligand molecules), as well as their structural diversity, the photoluminescence of MOFs can arise from a variety of possibilities [20]: (i) luminescence from organic linkers, which are normally extended π-conjugation systems with rigid structures, such as pyrene, anthracene, and their derivatives; (ii) metal-based emissions, e.g., MOFs with metal centers of lanthanoid; (iii) a metal–to–ligand charge transfer (MLCT), e.g., d^10^ Cu(I)- and Ag(I)-based MOFs; (iv) a ligand–to–metal charge transfer (LMCT), e.g., Zn(II)/Cd(II) and carboxylate ligand based MOFs; (v) antennae effects; and (vi) sensitization, e.g., MOFs with absorbing ligands and emitting lanthanoid ions (Figure 1). In the past, the competent luminescence features of MOFs have been extensively studied and exploited in fluorescence imaging and sensing applications [24]. These developed MOF-based luminescent probes can be classified into two types according to the recognition mechanism between the probe with the target analyte: (1) an MOF-based chemosensor, in which the analyte coordinates or interacts with the probe in a reversible manner, such as via physical/electrostatic interaction; (2) an MOF-based chemodosimeter, in which the target analyte can irreversibly react with a probe to yield a product that is chemically different from the starting probe via target-induced oxidation, hydrolysis, and nucleophilic processes.

MOF-based luminescent chemosensors have shown great potential in the detection of various analytes, including volatile organic compounds [25,26,27], ions (especially for metal cations) [28,29,30,31], gases [32,33,34,35], as well as for monitoring pH [36,37,38,39,40,41,42], humidity [43,44,45], and temperature [46,47,48]. Compared to the relatively well-developed fluorescent MOFs chemosensors, MOF-based chemodosimeters are expected to be more efficient in terms of selectivity, as they exploit the specific reactivities of certain target analytes [24]. Moreover, numerous MOF-based luminescent chemodosimeters can be afforded by introducing various recognition moieties for different analytes into building blocks. In the past several years, considerable effort has been expended in this field, and research interest is still growing. In spite of this interest, to date, the studies concerning MOF-based chemodosimeters have not yet been comprehensively addressed. To fill this gap, in the present review, we will make an effort to summarize recent progress in the development of MOF-based chemodosimeters. Moreover, the promising prospects and remaining challenges for future research in the field will also be discussed.

## 2. MOF-Based Luminescent Chemodosimeters for Sulfur Compounds

### 2.1. MOF-Based Chemodosimeters for H_2_S

Hydrogen sulfide (H_2_S), the smallest sulfhydryl compound, exists as a typical rotten egg smelling gas in the air or as a hydrosulfide ion while being dissolved in an aqueous solution under neutral pH conditions. Traditionally, H_2_S was simply considered to be an environmentally toxic species. In recent years, H_2_S has been discovered to be an essential biological molecule that can function as a cytoprotectant and gasotransmitter in organisms [49,50]. Moreover, the metabolism of H_2_S is closely related to various physiological and pathological events in the human body [51,52]. In this context, the development of new methods for sensing H_2_S and studying its biological roles has attracted tremendous attention [53,54,55]. Fluorescence technology is particularly promising for such purposes, as it enables monitoring of the target with superior temporal and spatial resolution [56,57,58,59], especially for in vivo applications. By exploiting its unique characteristics, such as strong nucleophilicity and ability to reduce potency, various small-molecule fluorescent probes have been developed for detecting and imaging H_2_S. These probes can be mainly categorized into three types according to their sensing strategy: (i) sensors based on the H_2_S-mediated reduction of an azide/nitro group to the amine [60,61,62,63]; (ii) sensors based on the H_2_S-participated nucleophilic reaction [64,65,66,67]; and (iii) sensors based on the binding reaction between sulfide and Cu^2+^ [68,69]. By exploiting the similar sensing strategies and carefully employing post-synthetic modification approaches, a number of MOF-based fluorescent H_2_S probes (Table 1) were successfully constructed in the last few years [70].

#### 2.1.1. Based on the H_2_S-Mediated Reduction of Azide/Nitro Group to Amine

Ghosh et al. first reported a Zr-based MOF, Zr_6_O_4_(OH)_4_(BDC-N_3_)_6_ (UiO-66@N_3_), bearing an azide group for sensing H_2_S [71]. The MOF-based chemodosimeter UiO-66@N_3_ was prepared via post modification of the amine functionalized UiO-66@NH_2_ using an azidation agent. UiO-66@N_3_ is highly stable in an aqueous solution and fluoresces very weakly due to the presence of an electron-deficient azide group (Figure 2). Treating UiO-66@N_3_ with Na_2_S in an HEPES (4-(2-hydroxyethyl)-1-piperazineethanesulfonic acid) aqueous buffer (10 mM, pH 7.4) creates a strong emission (16-fold enhancement), which can be ascribed to the target-mediated reduction of azide to amine, which produces the luminescent UiO-66@NH_2_. This conversion process is characterized by FTIR (Fourier Transform infrared spectroscopy) and NMR (Nuclear Magnetic Resonance Spectroscopy) studies. UiO-66@N_3_ can respond rapidly to H_2_S (less than 180 s) with high selectivity over other interferences, including most abundant biothiols (Cys and GSH). Moreover, UiO-66@N_3_ also displays low cell viability and has been applied to live cell imaging studies. The same research group subsequently prepared a nitro-functionalized Zr-MOF (UiO-66@NO_2_) for the fluorescence turn-on detection of H_2_S, in which UiO-66@NO_2_ can be facilely obtained in a single synthetic step by using 2-nitroterephthalic acid as the ligand [72]. Incorporation of the azide/nitro group onto other MOFs scaffolds has afforded a variety of luminescent H_2_S chemodosimeters. Qian et al. reported an azide-appended Zn-MOF for the fluorescent turn-on detection of H_2_S [73]. In a HEPES ethanol buffer, the probe displayed excellent selectivity and a rapid response time for H_2_S. By directly reacting the cerium(IV) nitrate with an azide/nitro-substituted 1,4-benzenedicarboxylate (BDC), Biswas et al. synthesized two Ce-MOFs (Ce-UiO-66@N_3_, Ce-UiO-66@NO_2_) for sensing H_2_S [74]. Since then, the Biswas group has developed a number of reaction-based luminescent H_2_S sensors, including Al(OH)(IPA-N_3_) [75], Zr_6_O_4_(OH)_4_((NDC-(NO_2_)_2_)_6_ [76], Al(OH)(BDC-N_3_) [77], and Zr_6_O_4_(OH)_4_(BDC-(NO_2_)_2_)_6_ [78]. Further, Al(OH)(BDC-N_3_) was also employed as a turn-off fluorimetric sensor for detecting Fe(III) in an aqueous solution [77].

Canivet et al. reported an Al-MOF, Al_3_(O)(OH)(BDC-N_3_)_3_ (Al-MIL-101-N_3_, Figure 3a) based sensing system for H_2_S [79]. It is worth noting that the use of a femtosecond (fs)-pulse laser excitation can significantly improve the emission features as well as the analytical performances of this MOF-based luminescent assay. All these investigated BDC-based MOFs (Al-MIL-101-NH_2_, In-MIL-68-NH_2_, and Zr-UiO-66-NH_2_) displayed characteristic emissions at about 460 nm (Figure 3b) under UV-lamp excitation at 343 nm. While under fs-pulse laser excitation, Al-MIL-101-NH_2_ revealed another predominent emission band centered at 565 nm, which may have resulted from an increase of the luminescence center in the electronic excited state and the redistribution of the photoexcited charge carriers (Figure 3c,d). The spectral behaviors of Al-MIL-101-N_3_ for H_2_S were studied in DMSO (dimethyl sulfoxide) as well as in HBSS (Hank’s balanced salt solution). In the biological media of HBSS, Al-MIL-101-N_3_ displayed a wide dynamic range of 0.1–120 μM and a low LOD of 100 nM for H_2_S (Figure 3e,f). Moreover, Al-MIL-101-N_3_ was applied to detect sulfide from an exogenous small molecule releaser (GYY4137) and endogenous H_2_S produced by 3T3L1 cells (Figure 3g,h).

By integrating Al-MOF (Al(OH)(BDC-NO_2_), Al-MIL-53-NO_2_), and poly(vinylidene fluoride) (PVDF), Qian et al. developed a novel MMM (MOF-polymer mixed-matrix membrane) for the fluorescence turn-on detection of H_2_S [80]. The Al-MIL-53-NO_2_@PVDF MMM can be readily prepared with a high loading of the MOF probes (70 wt%), following the procedures as depicted in Figure 4a. The obtained Al-MIL-53-NO_2_ MMM is mechanically robust and flexible and can be easily handled. The Al-MIL-53-NO_2_ MMM can be effectively applied to the flow-through detection of H_2_S, which exhibited remarkably high sensitivity with an LOD of 92.31 nM (Figure 4b,c).

Furthermore, the constructed Al-MIL-53-NO_2_ MMM was also used to detect H_2_S in real water samples.

#### 2.1.2. Based on the Binding Reaction between S^2−^ and Cu^2+^

Tang et al. firstly reported a Cu(II)-metalated MOF, Cu(TCPP)[AlOH]_2_ (TCPP: *meso*-tetrakis(4-carboxylphenyl)porphyrin), for luminescent sensing of H_2_S [81]. Cu(TCPP)[AlOH]_2_ was obtained via Cu(II) metalation of the porphyrin ring-contained parent MOF TCPP[AlOH]_2_(DMF)_3_(H_2_O)_2_. The paramagnetic Cu(II) ions can quench the ligand-based (porphyrin) fluorescence of Cu(TCPP)[AlOH]_2_. Upon the addition of H_2_S, the Cu(II) ion can be taken from the porphyrin center via the formation of a CuS precipitate, thereby leading to the recovery of porphyrin-based emissions in the MOF system (Figure 5a). The fluorescence response of Cu(TCPP)[AlOH]_2_ to H_2_S can occur instantaneously with high sensitivity (LOD: 16 nM) and excellent selectivity as other analytes did not generate any fluorescence enhancement. Luminescence imaging studies also demonstrated that Cu(TCPP)[AlOH]_2_ can be employed for monitoring both exogenous and endogenic H_2_S in liver hepatocellular (HepG2) cells (Figure 5b). The Tang group recently developed a CuO NPs functionalized NMOF (Nanoscale metal–organic frameworks) hybrid nanoprobe, CuO@TO@UiO-66, for the sensing and imaging of H_2_S [82]. Due to the energy transfer process from TO@UiO-66 to CuO NPs, the nanoprobe is weakly emissive. Sulfide can react with CuO to release the luminescent TO@UiO-66, thereby achieving a sensitive turn-on fluorescence response to H_2_S.

Qian et al. prepared a nano MOF Eu^3+^/Cu^2+^@UiO-66-(COOH)_2_ system for the ratiometric luminescent sensing of H_2_S by the post-modification of UiO-66-(COOH)_2_ with Eu^3+^ and Cu^2+^ ions [83]. This MOF system displayed two distinct emissions, a sharp Eu^3+^ emission at 615 nm and a broad ligand-based emission at 393 nm. Due to the decreased antenna efficiency of the H_4_btec ligands to Eu^3+^ in the presence of Cu^2+^, Eu^3+^/Cu^2+^@UiO-66-(COOH)_2_ exhibited a weak Eu^3+^ emission and a relatively enhanced ligand-based emission. While the target sulfide can effectively snatch the copper ion from the MOF probe and result in a significant enhancement in the fluorescence intensity ratio (I_615_/I_393_), thus achieving a ratiometric fluorescence response for H_2_S (Figure 6). Other advantageous features of this MOF sensor include excellent compatibility with aqueous media and instant response. However, it should be noted that other sulfhydryl compounds (i.e., GSH, Hcy, and Cys) also generated a certain degree of fluorescence response, which may be ascribed to the moderate binding affinity of H4btec to Cu^2+^, and these interferences also competitively bind to Cu^2+^. The same research group consequently developed a fluorescent MOF-based logic platform Eu^3+^/Ag^+^@UiO-66-(COOH)_2_ for H_2_S detection [84].

Yang et al. reported a luminescent composite Tb^3+^@Cu-MOF for the turn-on ratiometric sensing of H_2_S [85]. The synthesis process for Tb^3+^@Cu-MOF and the sensing mechanism for H_2_S are shown in Figure 7a. The Cu-MOF, [Cu(CPOC)_2_] was firstly prepared by reacting copper salt with an organic ligand (CPOC: 5-(4′-carboxyphenoxy) nicotinic acid). SC-XRD (single crystal X-ray diffraction) analysis indicated that [Cu(CPOC)_2_] belongs to the monoclinic system (P2_1_/c), with a half crystallographically independent metal center and one ligand per asymmetric unit (Figure 7b). The probe Tb^3+^@Cu-MOF can be readily obtained via post-grafting modification with Tb^3+^. [Cu(CPOC)_2_] displayed an emission band at about 390 nm. Tb^3+^@Cu-MOF showed additional emissions of the Tb^3+^ ion (at 489, 544, 585, and 620 nm) but with relatively weak intensities because the Cu^2+^ with an unsaturated electronic state (3d^9^) has a tendency to gain electrons and thus quench the fluorescence. With the addtion of sulfide, Cu^2+^ can be bounded and, as a consequence its quenching effect, can be hindered, leading to a significant and selective increase in the characteristic emissions of Tb^3+^ (Figure 7c). Noticeably, the Tb^3+^@Cu-MOF well retained its crystalline structure after being incubated with Na_2_S.

#### 2.1.3. Other MOF-Based Chemodosimeters for H_2_S

Cui et al. designed a vinyl-functionalized Zr-MOF (UiO-66-CH = CH_2_) for detecting H_2_S by using 2-vinylterephthalic acid as the linker ligand [86]. UiO-66-CH = CH_2_ showed a turn-off fluorescence response toward H_2_S with high sensitivity and selectivity. This probe also featured low toxicity and good water stability, but the sensing mechanism is not discussed in this study. Wang et al. reported a novel turn-on fluorescent assay for H_2_S based on the target-mediated collapse of an MOF structure (Fe_3_O(OH_2_)_3_(BDC-NH_2_)_3_, Fe^III^-MIL-88-NH_2_) (Figure 8) [87]. Due to the paramagnetic nature of Fe(III), Fe^III^-MIL-88-NH_2_ is non-emissive. Mixing the Fe^III^-MIL-88-NH_2_ suspension with a NaHS solution can lead to the breakdown of the MOFs and a release of the luminescent ligand of 2-aminoterephthalic acid.

### 2.2. MOF-Based Chemodosimeters for Biothiols

Biothiols such as Cys (cysteine), Hcy (homocysteine), and GSH (glutathione) play vital roles in various physiological and pathological processes [88,89,90,91]. During the past decade, a huge number of fluorescent probes have been developed for sensing and imaging biothiols by exploiting the specific reactivities of sulfhydryl group and/or amino group [92,93,94,95,96,97,98], including nucleophilic addition to electron-deficient unsaturated bonds, conjugate addition−cyclization reaction, cleavage of sulfonamide and sulfonate ester, cleavage of disulfide, and displacement of coordination to the metal complex. MOF-based Chemodosimeters for fluorescent sensing biothiols have emerged in the last two years.

Ghosh et al. reported a turn-on fluorescent probe, UiO-66-DNS, for selective sensing of biothiols [99]. UiO-66-DNS was prepared by post-grafting 2,4-dinitrosulfonyl moiety (DNS) to the chemically stable UiO-66-NH_2_ MOF. UiO-66-DNS exhibited weak fluorescence due to the PET (photoinduced electron transfer) from the ligand of 2-aminoterephthalic acid to the highly electron-withdrawing functional group of DNP (Figure 9a). After the addition of Cys to a UiO-66-DNS dispersed water solution (Figure 9b), a significant enhancement (ca. ∼48-fold) in luminescent intensity at 432 nm was obtained, which can be ascribed to the thiol-mediated cleavage of the DNS moiety and the release of UiO-66-NH_2_. Compared with Cys, UiO-66-DNS displayed slower response toward GSH because of its intrinsic bulkier feature for diffusing and interacting with the probe. Other amino acids did not generate any obvious enhancement in the emission intensity of the probe (Figure 9c), which demonstrated that the UiO-66-DNS is highly specific to biothiol.

Wu et al. developed a novel aldehyde-functionalized MOF, Cd–PPCA (Figure 10a), for selective sensing of Hcy [100]. The Cd–PPCA consisted of a Cd(II) metal center and two types of ligands, H_3_tca (4,4′,4″-tricarboxyltriphenylamine) and ppca (1H-pyrrolo-[2,3-*b*] pyridine-2-carbaldehyde). XRD (X-ray diffraction) results indicated that Cd–PPCA crystallized in the orthorhombic space group Pbnm with a = 25.500(5), b = 20.600(4), c = 13.700(3) Å (Figure 10b). Each unit building of trinuclear [Cd_3_(COO)_8_] contained one ppca ligand, with a nitrogen atom to coordinate with Cd and a desired aldehyde functional group for the specific recognition of the target Hcy. Suspended in a HEPES (pH = 7.4) buffer solution, Cd–PPCA showed a weakened fluorescence emission at about 450 nm compared with similar MOFs comprised of H_3_tca ligands and Cd^2+^ nods, which can be attributed to the PET from the triphenylamine groups to the electron withdrawing aldehyde moieties of the ppca ligand. With the addition of Hcy, the emission intensity of the assay showed a significant enhancement (Figure 10c), which can be ascribed to the selective reaction of the aldehyde moiety with Hcy and thus the inhibition of the PET process. This fluorescent response was also observed to be very fast (60 s) and sensitive (LOD: 40 nM). Moreover, the probe Cd–PPCA exhibited excellent specificity for Hcy over other species, including Cys (Figure 10d).

Wang and Zhang et al. developed a novel Zr-MOF based fluorescent PET switch/sensor, UiO-68-An/Ma [101]. The probe contains two kinds of ligands, one embedded with the anthracene unit serving as the luminophore and the other appended with a maleimide moiety as the PET acceptor (Figure 11a). The multivariate UiO-68-An/Ma favors a pseudo-PET process and only shows very weak fluorescence with an absolute PLQY (photoluminescence quantum yield) of 1.1%. Notably, the fluorescence behavior of UiO-68-An/Ma can be tuned by altering the acceptor moiety and thus the PET process. The authors firstly confirmed the tunable fluorescence response of UiO-68-An/Ma via a reversible D–A reaction with 3-furanmethanol. Furthermore, UiO-68-An/Ma was applied for sensing biothiols based on the well-established maleimide–thiol addition reaction. As a solid-state fluorescent turn-on sensor, UiO-68-An/Ma can sensitively response to biothiols (Cys, Hcy, and GSH) as low as 50 µM (Figure 11b).

With the in-situ encapsulation of rhodamine B (RhB) into Cu-BTC, Gao and Huang et al. reported a turn-on fluorescent assay (RhB@Cu-BTC MOFs) for sensing Cys [102]. The fluorescence of RhB@Cu-BTC was very weak because the embedded RhB was adjacent to the paramagnetic copper center. In the presence of Cys, the Cu-BTC framework collapsed, which resulted the release of RhB and thus a turn-on fluorescence response (Figure 12).

The MOF based fluorescent probe has also been exploited for sensing sulfur dioxide (SO_2_) or sulfite/bisulfite, another kind of important sulfur-containing specie on both biological and environmental aspects. Ghosh et al. constructed a MOF based luminescent probe, NH_2_-MIL-68(In)@CHO, for the monitoring of bisulfite [103]. The probe was prepared by the post synthetic approach via condensation of NH_2_-MIL-68(In) (In(OH)(bdc-NH_2_)) with glyoxal (Figure 13). The introduced aldehyde moiety can react with bisulfite to generate an OH group, which is available to form an intramolecular hydrogen bond and thus resulted in the inhibition of the C = N isomerization and the recovery of the fuorescence of the probe. Cui and Qian et al. prepared a Eu-BDC-NH_2_ film on the UiO-66-NH_2_ modified glass through an in situ secondary growth and successfully applied this functional film for sensing gaseous sulfur dioxide [104].

## 3. MOF-Based Chemodosimeters for Other Redox-Active Biomolecules

### 3.1. MOF-Based Chemodosimeters for HClO

Hypochlorous acid (HClO) is an important chemical reagent with wide application in various areas of organic synthesis, the cosmetics industry, food service, water treatment, etc. In living organisms, HClO is a kind of essential reactive oxygen species (ROS), which is normally produced in phagosomes via the reaction between hydrogen peroxide and chloride ions catalyzed by myeloperoxidase [105]. As a powerful oxidizer, the endogenous HClO can act as effective microbicidal agent when the host is being invaded by microbials [106]. On ther hand, HClO can also react with functioning biomolecules, such as proteins, nucleic acid, and fatty acids, which would produce adverse effects for organisms and correlate to numerous human diseases, such as kidney disease, cardiovascular diseases, and even cancers [107]. Therefore, the development of reliable analytical methods for monitoring HClO has attracted a great deal of attention [108,109]. Based on the unique characteristics of HClO, acting as both a potent oxidant and a good chlorination agent, various reaction-based organic fluorescent probes has been reported for selective sensing of HClO [110,111,112,113,114,115,116,117]. Until now, several examples of MOF-based chemodosimeters for luminescent detection of HClO have been presented.

Ma and Wang et al. developed the first MOF-based chemodosimetric probe for sensing HClO [118]. The designing strategy for the probe, UiO-Eu-L1 (L1: dimethyl 4-(carbaldehyde oxime) pyridine-2,6-dicarboxylate), is depicted in Figure 14. Uio-Eu-L1 was obtained by successively treating the Zr-MOF UiO-67 (Zr_6_O_4_(OH)_4_(BPDC)_6_) with europium ions and the functional ligand L1. Due to the efficient C = N isomerization-induced fluorescence quenching, UiO-Eu-L1 displayed very weak red emissions derived from europium ions. In the presence of HClO, the hydroxylamine moiety can be converted to aldehyde, which, in turn, leads to the inhibition of the C = N isomerization and thus the turn-on fluorescence response of the probe system. Ascribed to the long-lived phosphorescence of the Uio-Eu based MOF, this luminescent assay can efficiently eliminate the background signals and auto-fluorescence effects by the use of time-gated measurements. Uio-Eu-L1 also exhibited high sensitivity for HClO with a dynamic range of 0.1–5 μM and a detection limit of 16 nM.

Gu et al. presented a novel luminophore integrated MOF system, AF@MOF-801, for specific detection of HClO [119]. The composite AF@MOF-801 can be readily prepared via a one-step process by using 5-aminofluorescein (AF) as a co-reactant in the synthesis of MOF-801 (Figure 15a). AF can serve as a sensitive turn-off fluorescent HClO probe, as HClO can react with AF to produce chlorinated products. However, other coexistent biological molecules (such as dopamine, DA) also can lead to the similar fluorescent changes of AF. In the constructed sensory platform of AF@MOF-801, AF can be confined in the cages of the framework and the target HClO can diffuse into the framework to react with the probe AF, while the ultra-small aperture can block the entry of large-sized interferents. Due to this size-selective effect, AF@MOF-801 displayed excellent specificity for HClO. Only ClO^−^ produced the significant luminescence response of AF@MOF-801 (Figure 15b), while both ClO^−^ and DA generated strong fluorescence quenching for the free AF probe (Figure 15c). The feasibility of AF@MOF-801 for monitoring intracellular HClO was also demonstrated by MTT assay and flow cytometry analysis (FCA), as well as confocal laser scanning microscopy (CLSM) measurements (Figure 15d).

Chen et al. constructed a fluorophores@MOF (F1-Rubpy@ZnMOF74) nanocomposite-based ratiometric fluorescent probe for HClO by simultaneously encapsulating two fluorophores, fluorescein o-acrylate (F1) and tris(2,2′-bipyridyl)-dichroruthe-nium(II) hexahydrate (Rubpy), into ZnMOF74 [120]. F1-Rubpy@ZnMOF74 displayed two distinct emissions at 512 nm and 600 nm, corresponding to F1 and Rubpy, respectively. In the presence of HClO, the fluorescence of the target-responsive fluorophore F1 can be quenched, while the reference signal that originated from Rubpy remained unchanged, thus achieving a ratiometric response for HClO. In an aqueous HEPES buffer (pH = 7.5) solution, the fluorescence intensity ratio *(I*_512_*/I*_600_) of the F1-Rubpy@ZnMOF74 was found to be linearly correlated with the concentration of ClO^−^ with a dynamic range of 3.6 nM-100 μM.

### 3.2. MOF-Based Chemodosimeters for Ascorbic Acid

Chen and Qian et. al. developed a Ce-MOF sensor ZJU-136-Ce, (Me_2_NH_2_)_0.6_{[Ce^IV^(TPTC)]_0.4_-[Ce^III^(TPTC)]_0.6_}(H_2_O)_2_(H_4_TPTC = 1,1′:4′,1″-terphenyl-2′,4,4″,5′-tetracarboxylic acid) for sensing ascorbic acid (AA) [121]. Ce^IV^ in the probe ZJU-136-Ce can react with the AA to generate Ce^III^ and oxidized AA (DHA, dehydroascorbic acid). ZJU-136-Ce displays a luminescence band at 380 nm with a lifetime of 0.84 ns, which corresponds to the emissions of the Ce ion (Figure 16a). After reacting with AA, the probe system shows an increased emission band at about 400 nm (Figure 16b), which corresponds to the emission of the ligand. This fluorescence spectral response can be ascribed to the enhanced conjugation effect of the oxidized product DHA and thus the inhabited PET process from the TPTC ligand to the Ce ion. Subsequently, by introducing the Eu ions into the MOFs ZJU-136-Ce, the Qian group constructed a dual-emissive MOFs, ZJU-136-Ce_1−x_Eu_x_ (x = 0.24, 0.36), which can serve as an efficient ratiometric probe for monitoring AA [122]. The same research group also reported a new Zn-MOF, ZnL(H_2_O) (ZJU-137, H_2_L = 4,4′-(1H-pyrazole-1,3-diyl)dibenzoic acid, for the fluorescence “turn-off” detection of AA [123].

### 3.3. MOF-Based Chemodosimeter for 5-Hydroxytryptamine

Shi and Cheng et al. presented a Ln-MOF, Ln-MOF 1: {[Eu(TDA)(H_2_BTEC)_0.5_(H_2_O)_3_]·H_2_O}_n_ (H_2_TDA = thiazolidine 2,4-dicaboxylic acid, H_4_btec = 1,2,4,5-benzenetetracarboxylic acid), for sensing 5-hydroxytryptamine (HT) and 5-hydroxyindole-3-acetic acid (HIAA), which was synthesized by reacting Eu^3+^ with mixed ligands of H_2_TDA and H_4_BTEC [124]. Ln-MOF 1 is crystallized in the monoclinic space group P2_1_/c, in which each Eu^3+^ ion is nine-coordinates in a spherical capped square antiprism coordination geometry (Figure 17a). Due to the conjugated π system and the pronounced antenna effect of the ligand BTEC, Ln-MOF 1 displays intense characteristic emissions of the Eu^3+^ emitter with a maximum peak located at 616 nm, which is ascribed to the ^5^D_0_→^7^F_2_ transition of Eu^3+^ ion. In the presence of HT or HIAA, the luminescence of the Ln-MOF 1 can be effectively quenched (Figure 17b,c). This process was attributed to the competitive absorption of excitation light by the analyte and the ligand. Ln-MOF 1 displayed several favorable features for HT and HIAA sensing, including high stability over a wide pH range and long-term storage, excellent sensitivity, and fast response time.

### 3.4. MOF-Based Chemodosimeter for H_2_O_2_

The hydrogen peroxide (H_2_O_2_) mediated conversion of arylboronates to phenols has been widely exploited for the development of fluorescent H_2_O_2_ probes [125]. On the basis of this unique chemical reaction, Biswas et al. developed a Zr-MOF based probe, Zr-UiO-66-B(OH)_2_, for sensing H_2_O_2_ [126]. Zr-UiO-66-B(OH)_2_, which can be can be easily prepared by reacting ZrOCl_2_·8H_2_O with the linker of BDC-B(OH)_2_ (2-boronobenzene-1,4-dicarboxylic acid) (Figure 18). Zr-UiO-66-B(OH)_2_ can act as a sensitive and selective off−on luminescent chemodosimeter for H_2_O_2_ with a 4-fold increment in the fluorescence intensity upon the addition of an excess amount of the target. Moreover, the probe Zr-UiO-66-B(OH)_2_ was successfully applied to image H_2_O_2_ in MDAMB-231 cells.

## 4. MOF-Based Chemodosimeters for Ions

### 4.1. MOF-Based Chemodosimeters for Fluoride Ions

The fluoride ion, F^−^, as a typical hard Lewis base with the smallest ionic radius and highest charge density, has attracted much interest due to its association with various biological, medical, and technological processes. Commonly, fluoride is considered to be a critical component for preventing dental caries as fluoride can promote the formation of enamel-strengthening fluorapatite [127,128,129]. However, the over intake of fluoride can lead to excess mineralization within oganisms and cause gastric and kidney problems. Various analytical techniques have been developed for sensing fluoride [130], including the well-established electrochemical method [131,132], chromatography [133,134], and colorimetric and fluorogenic assays [135]. In the past decades, numerous fluorescent probes have been developed for fluoride based on different fluoride-participated processes, such as fluoride-induced Si−O or Si−C bond cleavage, H−F hydrogen bonding formation, and Lewis acid−base interactions [136,137,138]. By exploiting these special reactivities of fluoride, several MOF-based luminesscent probes have recently been developed for its detection [139,140].

Yin et al. reported a reaction-based MOF probe (Eu-bop, Eu_2_(isp)_3_(H_2_O)_2_) for fluoride ions [141]. Eu-bop was prepared by using Eu^3+^ as the metal node and 5-bop (5-boronoisophthalic acid) as the ligand that contains a recognition moiety of a boric acid group for the target F^−^. The substituted boric acid group also can tune the electronic structure of the ligand and resulted in an incomplete energy transfer from the ligand to Eu^3+^ emitter in the probe Eu-bop. Therefore, the Eu-bop displayed two emission bands: 366 nm, corresponding to the ligand of 5-bop; and 570–750 nm, corresponding to Eu^3+^. Upon the addition of fluoride ions, the emission intensity at 366 nm was significant increased with a concomitant decrease in emissions at 625 nm (Figure 19a). The fluorescence response can be ascribed to the OH/F exchange reaction on the boron atom, which changed the hybridization state of boron from sp^2^ (−B(OH)_2_) to sp^3^ (−BF_3_), and thus disrupted the p_π_−π conjugation of the 5-bop and the decreased intersystem crossing efficiency (Figure 19b), as well as the antenna effect of the ligand. The fluorescence intensity ratio (I_625_/I_366_) of the probe system was found to vary linearly with the concentration of fluoride in a range of 4–80 μM. The probe was also applied to detect fluoride in real samples of river and underground water.

Recently, Stylianou et al. developed a luminescent lanthanide MOF, ([Eu(tctb)(H_2_O)]·2DMF), referred to as SION-105 for the recognition of fluoride ion [142]. SION-105 consisted of tris (p-carboxylic acid) tridurylborane ligand (tctb^3−^) and an Eu^III^ metal center. The ligand tctb^3−^ contains a three-coordinate B acting as the recognition site for F^−^ and the surrounding duryl groups offering size-selective steric protection from other interferences (Figure 20a). SION-105 displayed a strong characteristic Eu^III^ luminescent emission due to the efficient antenna effect of the ligand tctb^3−^ (Figure 20d). Upon the addition of F^−^, the emissions of the probe can be quenched due to the specific interaction of F^−^ with the B Lewis acid site. A linear luminescence quenching response of SION-105 towards the F^−^ ion was observed in a range of 0.5 to 2.0 ppm (Figure 20b). A Stern−Volmer plot of quenching with the F^−^ concentration indicates the occurrence of both static and dynamic quenching (Figure 20c).

### 4.2. MOF-Based Chemodosimeter for Hg^2+^

Mercury is one of the most toxic heavy metals in the environment. In aqueous media, mercury can be transformed into methylmercury, a powerful neurotoxin, which can be accumulated and ingested by humans through the food chain [143,144,145]. In the human body, methylmercury can lead to serious symptoms, such as cognitive and motor disorders, neurological impairments, brain damage, and even death [146,147,148]. These environmental and biological problems have prompted the rapid development of techniques for sensing mercury [149,150,151,152,153]. Until present, many reaction-based fluorescent probes for Hg^2+^ have been developed based on various Hg^2+^-induced chemical processes [154,155,156], such as desulfation or deselenization processes, desulfation and cyclization processes, thiol elimination process, and the oxymercuration−elimination of vinyl/alkyne ether. Recently, Ghosh reported a MOF probe (UiO-66@Butyne) for the reaction-based detection of Hg^2+^ [157]. UiO-66@Butyne was prepared from Zr ion and a ligand of 2,5-bis (but-3-yn-1-yloxy) terephthalic acid, which contained a butyne moiety acting as the recognition site for Hg^2+^ (Figure 21a). The prepared UiO-66@Butyne retained the crystalline structure of MOF UiO-66 and showed a strong green emission (λ_em,max_ = 537 nm). Addition of Hg^2+^ to the aqueous solution of UiO-66@Butyne can result in fluorescent quenching of the system (Figure 21b), which can be attributed to the target-induced conversion of UiO-66@Butyne to the less fluorescent product of UiO-66@OH through the process of oxymercuration−elimination of ethynyl ether. As this MOF-based chemodosimeter exploited the specific reactivity of the target, UiO-66@OH exhibited excellent selectivity for Hg^2+^.

### 4.3. MOF-Based Chemodosimeter for CN^−^

The cyanide ion is an extremely poisonous chemical with widespread applications in industries such as metal mining, electroplating, and plastic and fertilizer manufacturing [158,159]. Various methods have been developed for the quantitative analysis of CN^−^, including the titration method, electrochemical assays [160], chromatography [161], as well as the colorimetric and fluorogenic method [162]. Among these methods, the design and use of fluorescent probes for sensing CN^−^ have received considerable attention [163,164,165]. The chemososimeters for CN^−^ monitoring are normally associated, with several typical chemical reactions, including the cyanohydrin forming process, additions to the dicyano-vinyl group, michael addition, and indolium or pyridinium addition reactions. Recently, based on the specific addtion reaction of CN^−^ dicyano-vinyl group, Ghosh developed a Zeolitic imidazolate framework, M-ZIF-90 ([Zn(C_8_H_6_N_4_O_2_)]_n_), for fluorescence sensing CN^−^ [166]. M-ZIF-90 was prepared by incorporating the recognition moiety of dicyano-vinyl onto the aldehyde-appended ZIF-90 via post-synthetic modification (Figure 22). In the H_2_O/DMSO (1:1) mixture, M-ZIF-90 displayed a turn-off fluorescence response towards CN^−^ based on the nucleophilic addition of CN^−^ to the dicyano-vinyl group, which would interrupt the π-conjugation of the probe and thus result in significant fluorescence change.

## 5. Conclusions

In recent years, the design, synthesis, and application of MOFs have attracted an ever-increasing interest of researchers in the fields of chemistry, physics, materials, and engineering. The porous and crystalline nature, versatile selections for metal nodes and organic linkers, and tunable structures also endow MOFs with unique characteristics for sensing applications. In this review, we summarized recently developed MOF-based luminescent chemodosimeters. These reaction-based luminescent probes inherit the robust structural features of MOFs (e.g., large surface area, high porosity, and good thermal and chemical stability) and also exhibit excellent selectivity as they exploit the specific reactivities of the target analyte. Several strategies have been involved in the development of MOF-based chemodosimeters: (i) installing the recognition moiety onto the ligand via a post-synthetic modification approach or using a recognition moiety-appended ligand for preparing MOFs; (ii) construction of small-molecule fluorescent probe@MOF hybrid nanocomposites, in which the MOFs act as a carrier to confine the responsive probe; (iii) based on the target-induced collapse of the MOF structure, etc. Further, the MOF scaffolds exploited for constructing luminescent chemodosimeters are normally required to possess several features, including high stability and dispersibility in aqueous (or aqueous-containing) media, feasibility of installing the recognition moiety for certain analytes, environmental friendliness, and/or good biocompatibility for biosensing and bioimaging applications. Until now, although, dozens of MOF-based chemodosimeters have been reported for different analytes with several desirable traits, such as high selectivity and sensitivity, good stability, and rapid response, some critical performance limitations for this type of luminescent probe still need to be improved, including short-wavelength light excitation and/or emission, insufficient designing strategies for the probe, and limited target analytes. To overcome these shortcomings, several future directions for the design of MOF-based chemodosimeters are proposed herein.

(i) Employing New Rationally Designed Ligands

Most of the developed MOF-based chemodosimeters are ligand-based emission systems. Currently, the used ligand normally belongs to a blue-emissive fluorophore (e.g., BDC) with ultraviolet to blue light excitation, which is unfavorable for applications in complex samples, as well as in bioimaging. Future efforts should focus on the design and utilization of new ligands for constructing luminescent MOFs that have typical spectral features, such as visible/near-infrared emission, two-photon excitation, and large Stokes shifts.

(ii) Developing Dual-Ligand Mofs Systems

Compared with the single-ligand system, the dual-ligand MOFs have greater structural and functional diversity. For example, a fluorescence resonance energy transfer (FRET) sensing system could be achieved by carefully designing the electronic structure of two different ligands. Ratiometric fluorescent probes could be obtained by using one ligand as the target-responsive unit and the other ligand as the internal signal reference.

(iii) Expanding the Application Scope for Mof-Based Chemodosimeters

To date, MOF-based chemodosimeters have only been employed for sensing some ions and small molecules, such as F^−^, Hg^2+^, CN^−^, H_2_S, HClO, Cys, H_2_O_2_, AA, and HT. It is anticipated that future efforts will focus on developing MOF-based chemodosimeters for monitoring other important biomolecules, such as bioenzymes, as well as for bioimaging applications.

In summary, MOF-based luminescent chemodosimeters have increasingly attracted research interest. It can be expected that ongoing studies in this area will lead to the rapid development of more effective luminescent nanomaterials for sensing and imaging applications.

## Figures and Tables

**Figure 1 nanomaterials-09-00974-f001:**
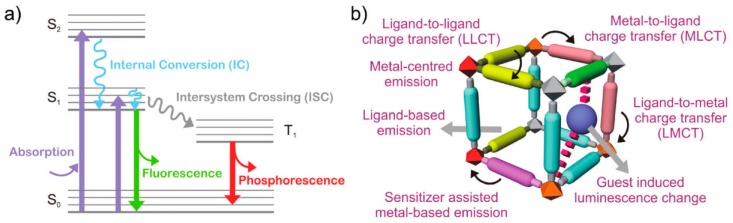
(**a**) Partial energy-level diagram for photoluminescent processes; (**b**) various possibilities for the emission of metal–organic frameworks (MOFs). Reproduced with permission from [24]. Copyright the Royal Society of Chemistry, 2017.

**Figure 2 nanomaterials-09-00974-f002:**
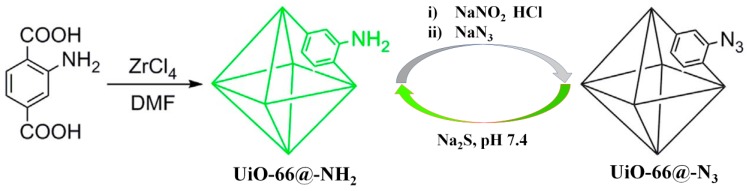
The synthetic route and sensing mechanism of UiO-66@N_3_ for H_2_S. Reproduced with permission from [71]. Copyright Nature Publishing Group, 2014.

**Figure 3 nanomaterials-09-00974-f003:**
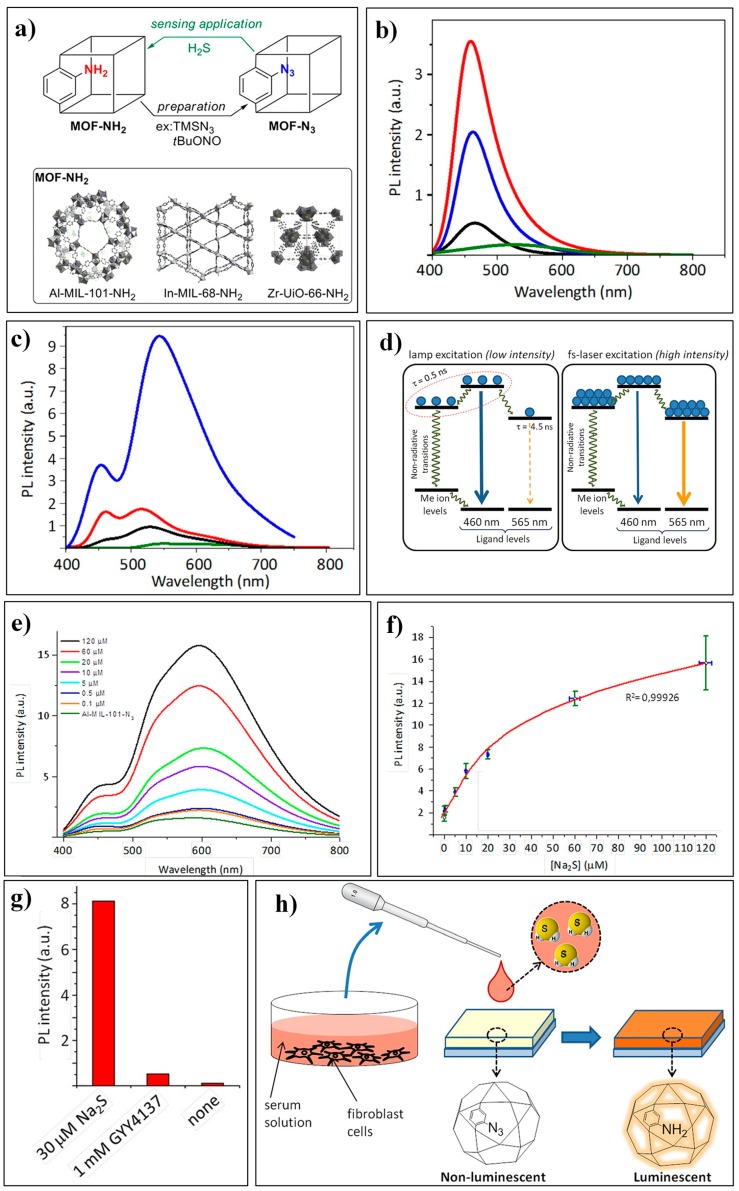
(**a**) Azido-MOFs for H_2_S sensing. (**b**,**c**) Emission spectra of dry MOF samples, Al-MIL-101-NH_2_ (blue), In-MIL-68-NH_2_ (red), Zr-UiO-66-NH_2_ (black), Al-MIL-101-N_3_ (green) under UV-lamp excitation (**b**) or fs-pulse laser excitation (**c**) at 343 nm. (**d**) Proposed schematic mechanisms of charges recombination underpinning the observed switch in emission channels. (**e**) Emission spectra of Al-MIL-101-N_3_ in the presence of different concentrations of sodium sulfide (excited at 343 nm with a fs-pulse laser). (**f**) Reversed cubic fit of sodium sulfide concentration vs. measured intensity of the 565 nm band in emission spectra. (**g**) Detection of H_2_S released from GYY4137. (**h**) Exposure of Al-MIL-101-N_3_ to a sample of culture medium for cells with endogenously produced H_2_S. Reproduced with permission from [79]. Copyright Wiley-VCH, 2016.

**Figure 4 nanomaterials-09-00974-f004:**
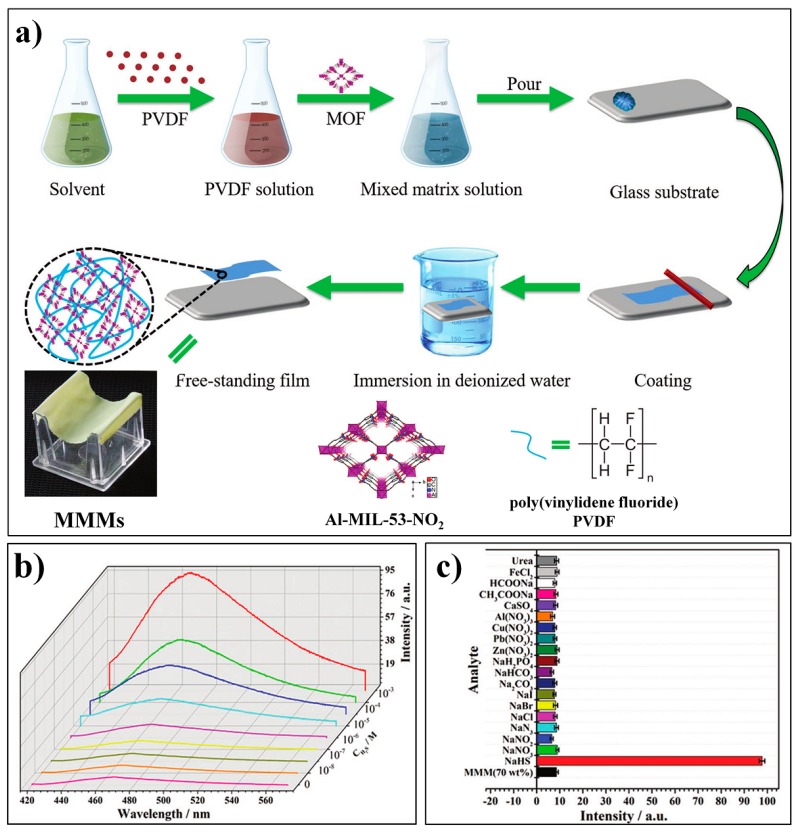
(**a**) Fabrication process for Al-MIL-53-NO_2_@PVDF MOF-polymer mixed-matrix membranes (MMMs). (**b**) Fluorescence spectra of Al-MIL-53-NO_2_@PVDF MMMs (70 wt%) with different concentrations of H_2_S (excited at 396 nm). (**c**) Fluorescence response of Al-MIL-53-NO_2_ MMMs at 466 nm toward various analytes. Reproduced with permission from [80]. Copyright Wiley-VCH, 2016.

**Figure 5 nanomaterials-09-00974-f005:**
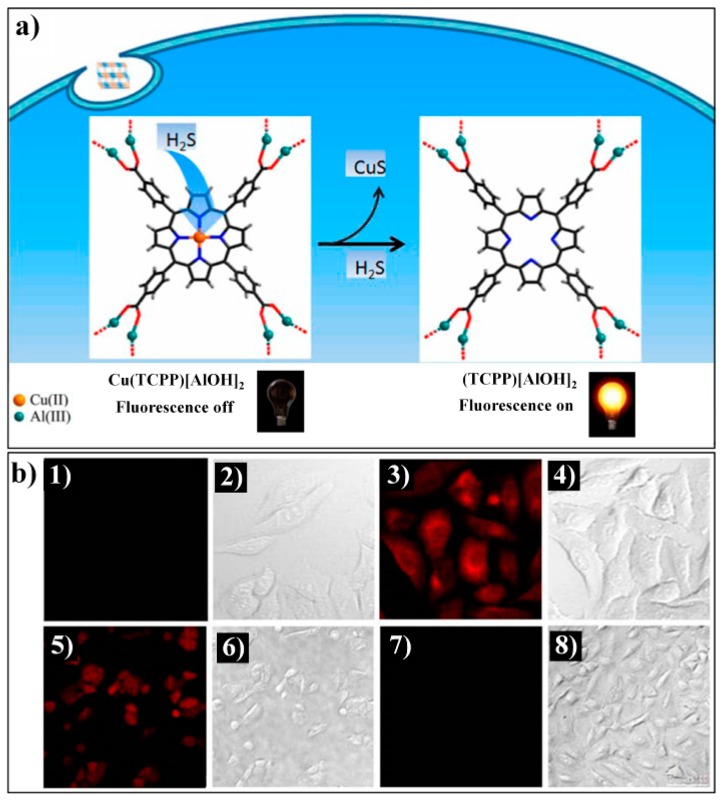
(**a**) The proposed strategy for the fluorescent variation of Cu(TCPP)[AlOH]_2_ for H_2_S. (**b**) Confocal fluorescence images in living cells: (1) image of HepG2 cells incubated with 10 μM Cu(TCPP)[AlOH]_2_; (2) bright-field image of (1); (3) image of HepG2 cells incubated with 10 μM Cu(TCPP)[AlOH]_2_ and 50 μM NaHS; (4) bright-field image of (3); (5) image of A549 cells incubated with 500 μM SNP and 10 μM Cu(TCPP)[AlOH]_2_; (6) bright-field image of (5); (7) image of A549 cells incubated with 250 mg·L^−^^1^
dl-propargylglycine (PPG) and 10 μM Cu(TCPP)[AlOH]_2_; (8) bright-field image of (7). Reproduced with permission from [81]. Copyright American Chemical Society, 2014.

**Figure 6 nanomaterials-09-00974-f006:**
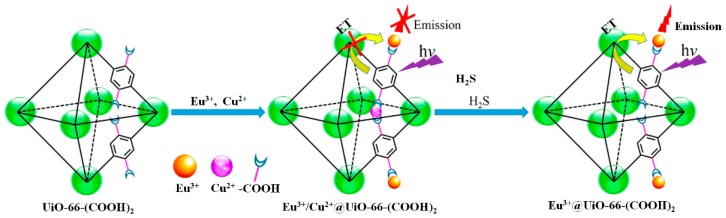
The fluorescence detection mechanism of the MOF Eu^3+^/Cu^2+^@UiO-66-(COOH)_2_ system for H_2_S. Reproduced with permission from [83]. Copyright American Chemical Society, 2016.

**Figure 7 nanomaterials-09-00974-f007:**
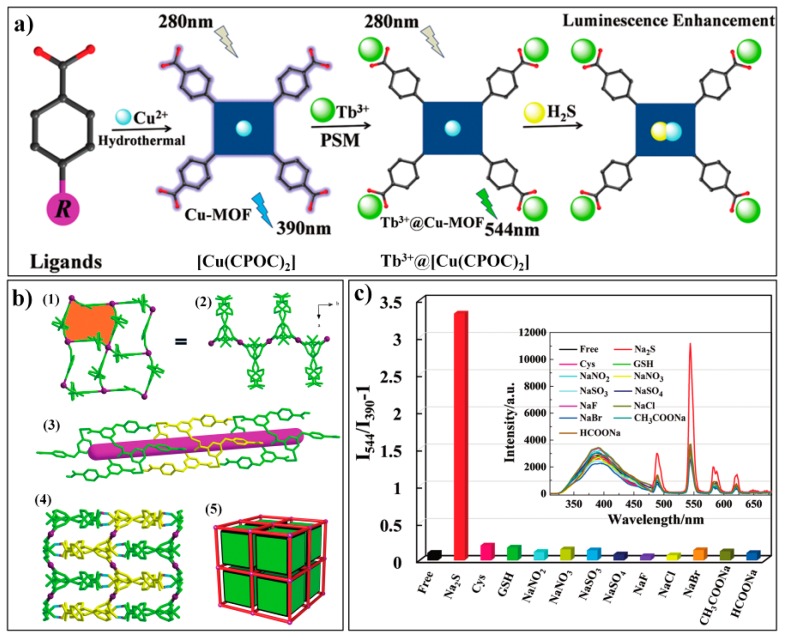
(**a**) Synthetic route for Tb^3+^@Cu-MOF and its sensing mechanism for H_2_S. (**b**) Crystalline structure of [Cu(CPOC)_2_], (1) 2D layers of Cu1; (2) 2D layers of Cu1 viewed along the c axis; (3) 1D channel in the 3D framework; (4) 3D supramolecular framework of [Cu(CPOC)_2_] through O–H···O interactions; (5) topological representation of the 3D structure. (**c**) Fluorescence response of Tb^3+^@[Cu(CPOC)_2_] towards sulfide and other analytes. Reproduced with permission from [85]. Copyright the Royal Society of Chemistry, 2017.

**Figure 8 nanomaterials-09-00974-f008:**
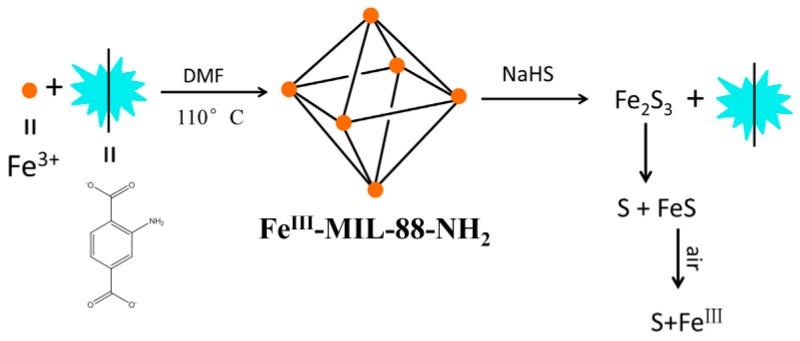
The synthesis and the turn-on fluoresence response of Fe^III^-MIL-88-NH_2_ for H_2_S. Reproduced with permission from [87]. Copyright Elsevier B.V., New York, NY, USA, 2017.

**Figure 9 nanomaterials-09-00974-f009:**
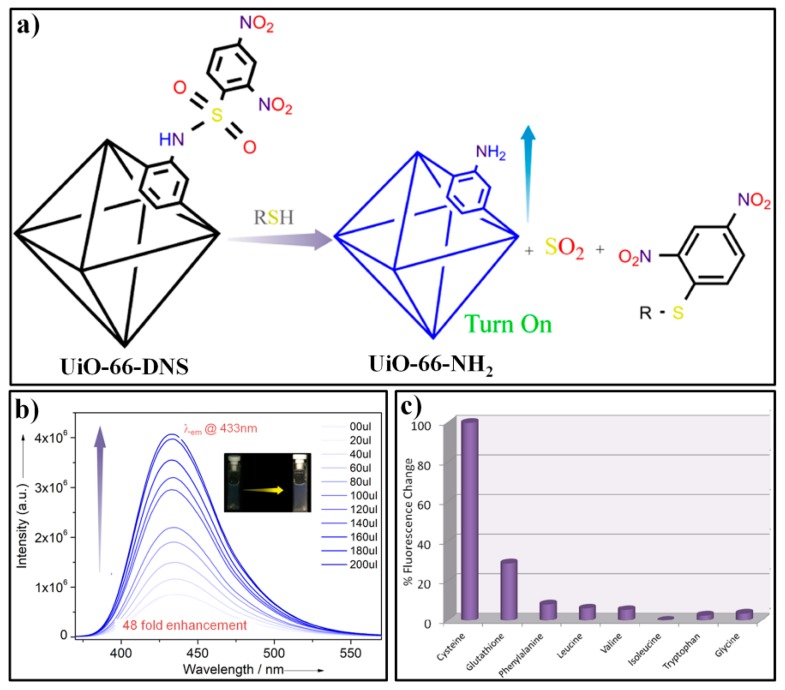
(**a**) The fluorescence detection mechanism of UiO-66-DNS for biothiols. (**b**) The fluorescence responses of UiO-66-DNS after addition of different amount of Cys. (**c**) Relative changes in the fluorescence intensity of UiO-66-DNS upon treatment with various amino acids. Reproduced with permission from [99]. Copyright American Chemical Society, 2016.

**Figure 10 nanomaterials-09-00974-f010:**
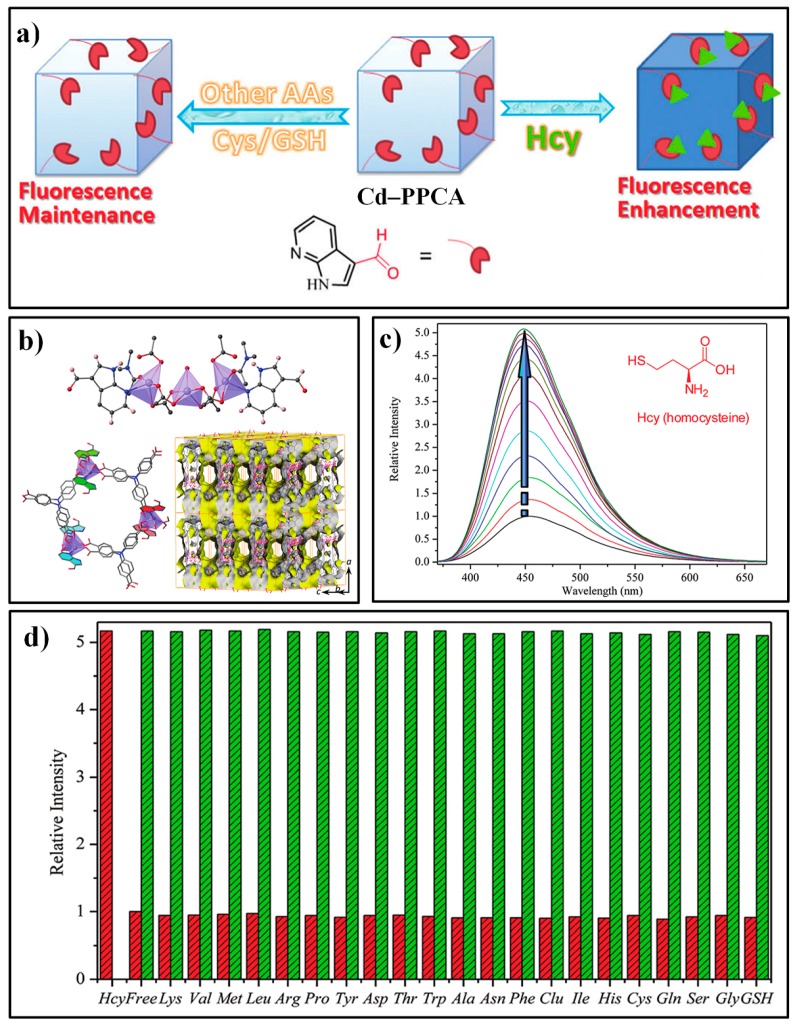
(**a**) The fluorescence detection mechanism of Cd–PPCA for Hcy. (**b**) the coordinated environment of the Cd^2+^ in Cd–PPCA (Top); a two-dimensional double-layer structure composed of Cd^2+^ and ligands of ppca and H_3_tca (bottom-left); the Connolly surface of the framework of Cd–PPCA (bottom-right). (**c**) The fluorescence spectra of Cd–PPCA upon addition of different amounts of Hcy. (**d**) Results for selectivity and competition tests. Reproduced with permission from [100]. Copyright the Royal Society of Chemistry, 2018.

**Figure 11 nanomaterials-09-00974-f011:**
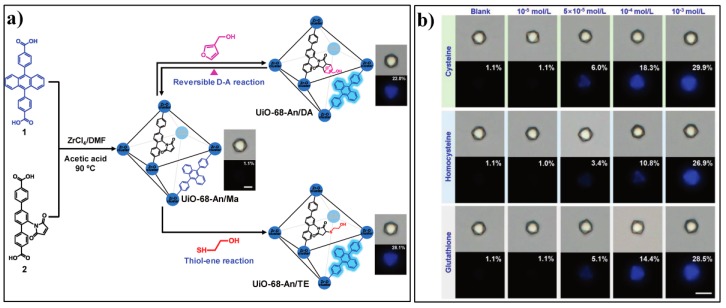
(**a**) Synthesis of UiO-68-An/Ma and tuning the fluorescent photoinduced electron transfer (PET) in the MOF through a reversible D–A reaction or thiol-ene reaction. (**b**) Bright-field and photoluminescence single-crystal images of UiO-68-An/Ma treated with different concentrations of Cys, Hcy, and GSH for 5 min. Reproduced with permission from [101]. Copyright Wiley-VCH, 2016.

**Figure 12 nanomaterials-09-00974-f012:**
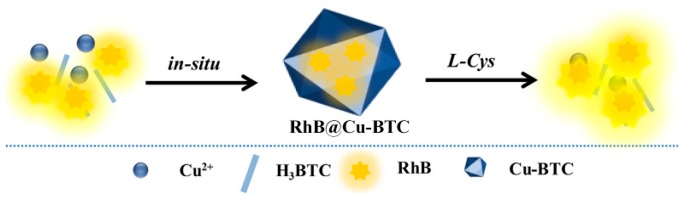
The synthesis of RhB@Cu-BTC and its sensing process for Cys. Reproduced with permission from [102]. Copyright 2018 Elsevier B.V., New York, NY, USA.

**Figure 13 nanomaterials-09-00974-f013:**
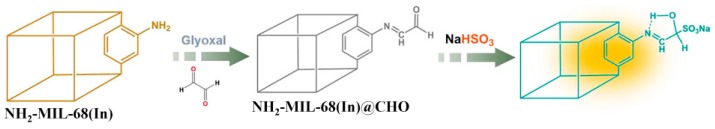
The synthesis of NH_2_-MIL-68(In)@CHO and its sensing process for bisulfite. Reproduced with permission from [103]. Copyright Elsevier B.V., New York, NY, USA, 2018.

**Figure 14 nanomaterials-09-00974-f014:**
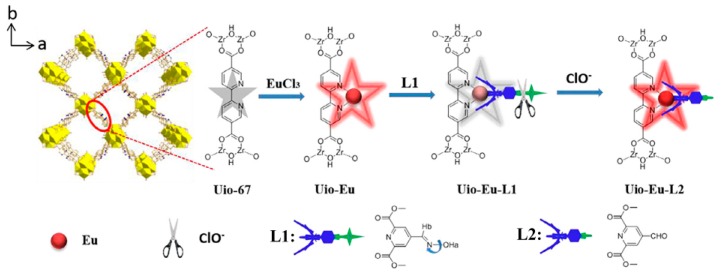
The structure and sensing mechanism of Uio-Eu-L1 for ClO^−^. Reproduced with permission from [118]. Copyright Elsevier B.V., New York, NY, USA, 2018.

**Figure 15 nanomaterials-09-00974-f015:**
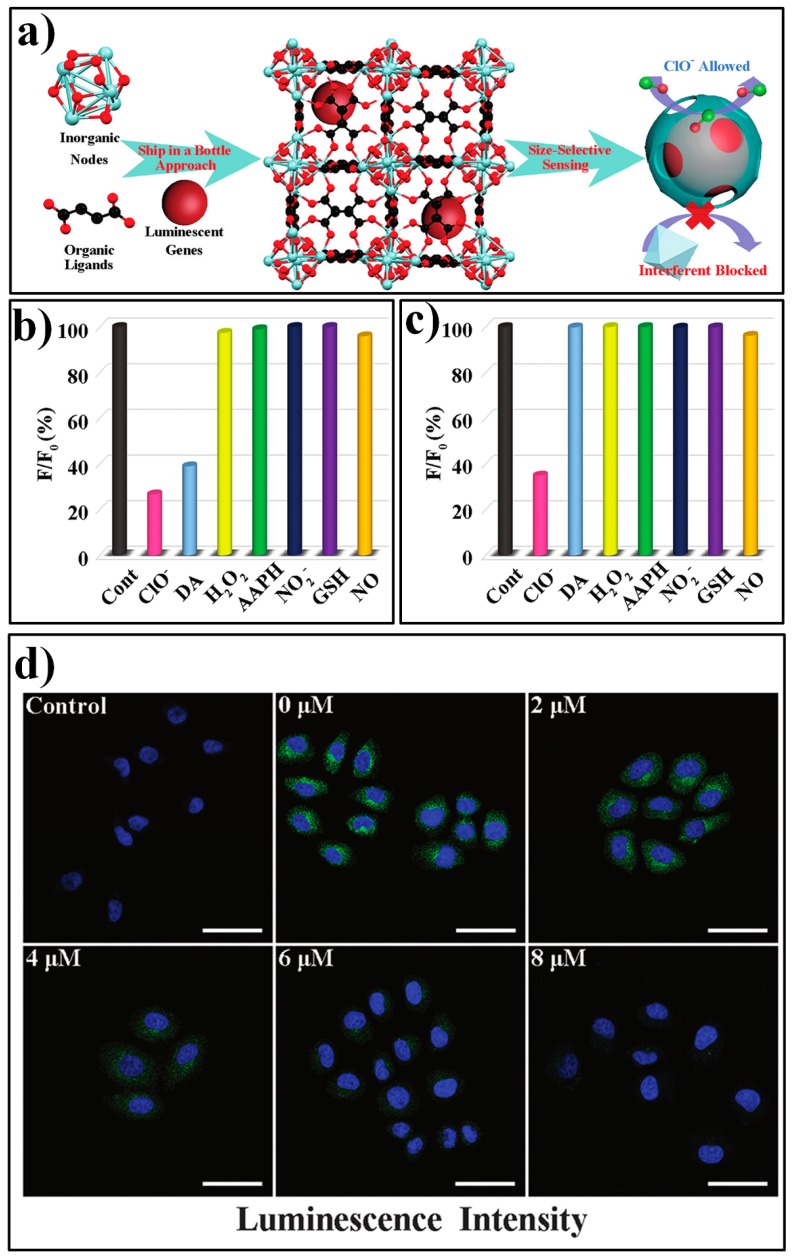
(**a**) Schematic illustration of the synthetic procedure for AF@MOF-801 and the size-selective sensing mechanism toward targe analyte of ClO^−^. (**b**) The luminescence responses of free AF for various analytes. (**c**) The luminescence responses of AF@MOF-801 for various analytes. (**d**) The confocal laser scanning microscopy images of SMMC-7721 cells incubated with AF@MOF-801 and with various concentrations of ClO^−^. Reproduced with permission from [119]. Copyright the Royal Society of Chemistry, 2019.

**Figure 16 nanomaterials-09-00974-f016:**
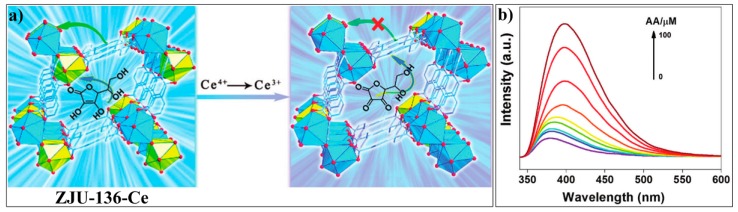
(**a**) Schematic illustration of the sensing mechanism of ZJU-136-Ce for asorbic acid (AA). (**b**) Luminescence spectra of ZJU-136-Ce in the presence of different concentrations of AA. Reproduced with permission from [121]. Copyright the Royal Society of Chemistry, 2017.

**Figure 17 nanomaterials-09-00974-f017:**
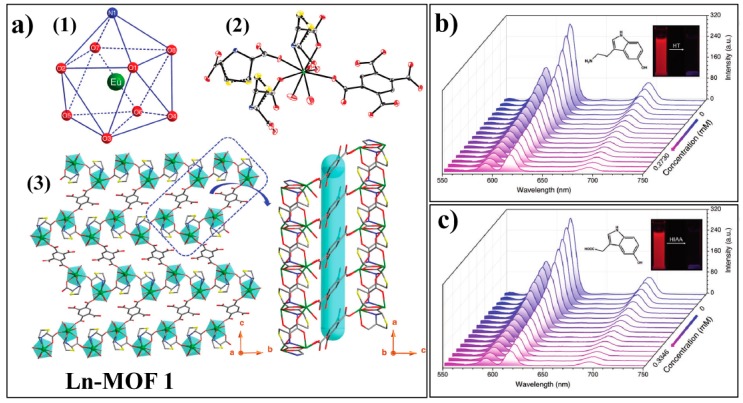
(**a**) structure of Ln-MOF 1: (1) Coordination geometry of the Eu^3+^; (2) The asymmetric unit of Ln-MOF 1; (3) 3D framework composed of Ln-MOF 1. Atom codes: Eu (green), C (gray), N (blue), O (red), and S (yellow). Emission spectra of Ln-MOF 1 upon addition increase amount of (**b**) HT and (**c**) HIAA. Reproduced with permission from [124]. Copyright Wiley-VCH, 2018.

**Figure 18 nanomaterials-09-00974-f018:**
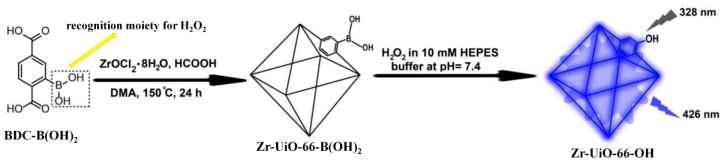
Schematic illustration of the sysnthesis of Zr-UiO-66-B(OH)_2_ and the sensing process for H_2_O_2_. Reproduced with permission from [126]. Copyright American Chemical Society, 2018.

**Figure 19 nanomaterials-09-00974-f019:**
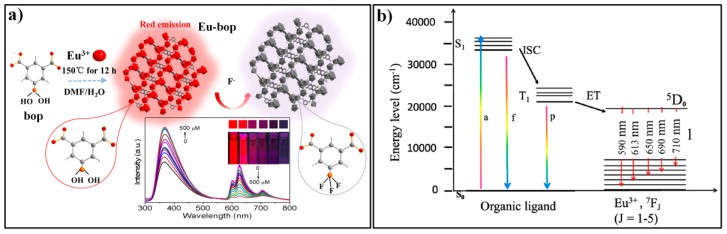
(**a**) The synthesis of Eu-bop and its sensing process for F^−^. (**b**) Schematic representation of absorption, migration, and emission of Eu-bop. Reproduced with permission from [141]. Copyright American Chemical Society, 2017.

**Figure 20 nanomaterials-09-00974-f020:**
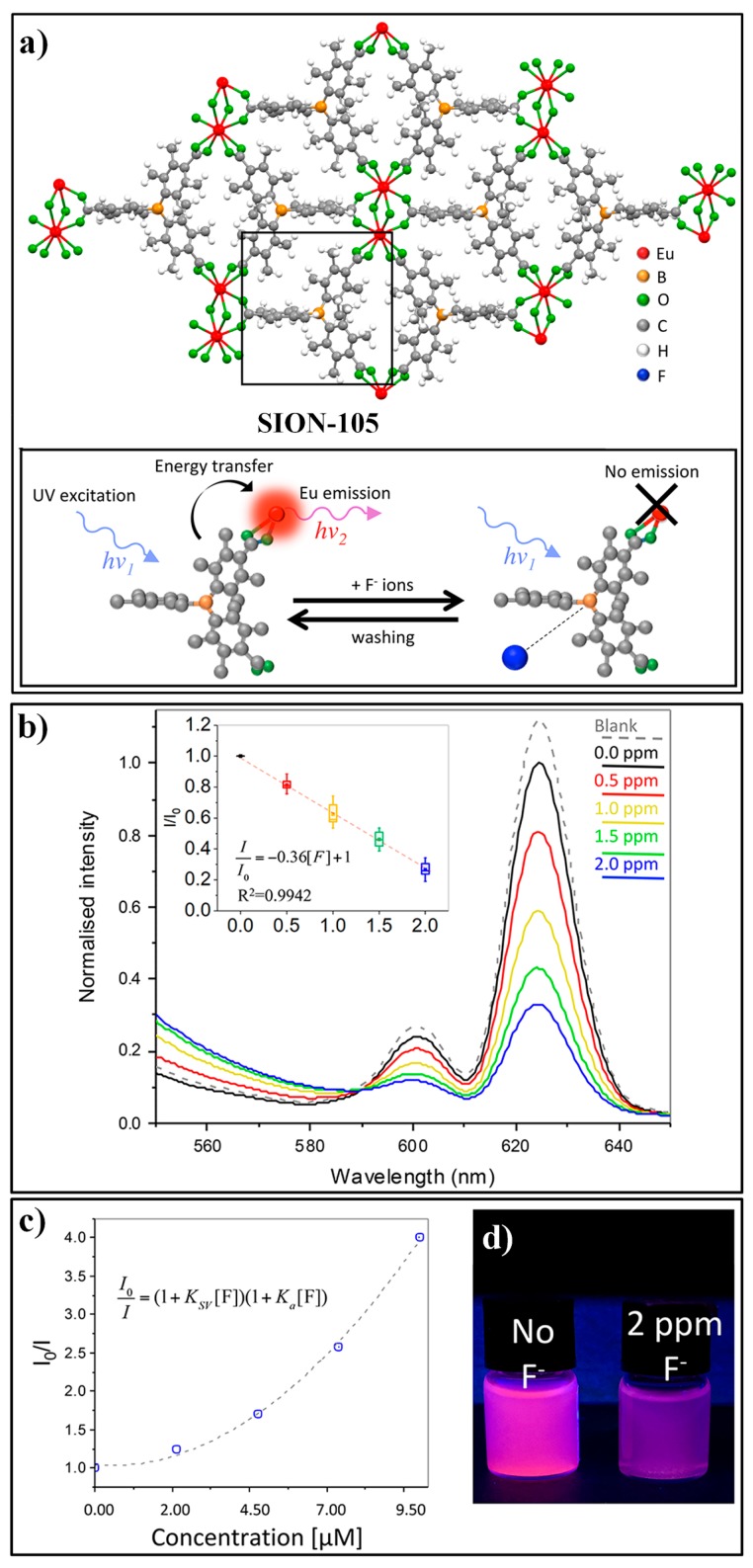
(**a**) The structure of SION-105 and its sensing process for F^−^. (**b**) Luminescence spectra of SION-105 upon the addition of F^−^ with different concentrations. (**c**) Stern−Volmer plot of quenching with increasing F^−^ concentration. (**d**) Photographs of SION-105 suspension with or without of F^−^. Reproduced with permission from [142]. Copyright American Chemical Society, 2019.

**Figure 21 nanomaterials-09-00974-f021:**
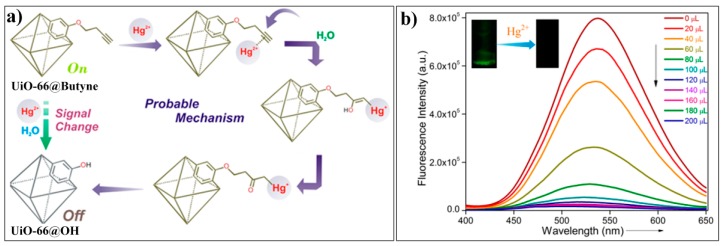
(**a**) Schematic illustration of the sensing mechanism of UiO-66@Butyne for Hg^2+^. (**b**) Fluorescence spectra of UiO-66@Butyne upon addition of different amount of Hg^2+^. Reproduced with permission from [157]. Copyright American Chemical Society, 2018.

**Figure 22 nanomaterials-09-00974-f022:**
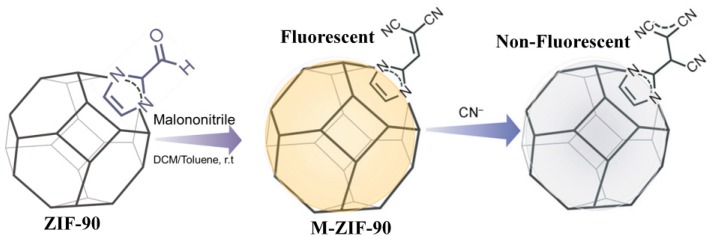
Schematic illustration of the the sysnthesis of M-ZIF-90 and the sensing process for CN^−^. Reproduced with permission from [166]. Copyright Wiley-VCH, 2016.

**Table 1 nanomaterials-09-00974-t001:** List of MOF-based luminescent chemodosimeters for H_2_S.

MOF Formula	λ_ex_/λ_em_ (nm)	Dynamic Range	LOD ^1^	RT ^2^	Media	Real Sampe	Ref.
Zr_6_O_4_(OH)_4_(BDC-N_3_)_6_	334/436	0–4 mM	117 μM	180 s	HEPES buffer (10 mM, pH 7.4)	Live cells	[71]
Zr_6_O_4_(OH)_4_(BDC-NO_2_)_6_	334/436	0–4 mM	188 μM	460 s	HEPES buffer (10 mM, pH 7.4)	--	[72]
Zn_4_O(OH)_4_(BDC-N_3_)_3_	395/455	0–0.5 mM	28.3 μM	90 s	HEPES ethanol buffer (10 mM, pH 7.4)	--	[73]
Ce_6_O_4_(OH)_4_(BDC-N_3_)_6_	334/429	0–3.5 mM	12.2 μM	760 s	HEPES buffer (10 mM, pH 7.4)	--	[74]
Ce_6_O_4_(OH)_4_(BDC-NO_2_)_6_	334/429	0–3.5 mM	34.8 μM	480 s	HEPES buffer (10 mM, pH 7.4)	--	[74]
Al(OH)(IPA-N_3_)	330/405	0–0.06 mM	2.65 μM	420 s	HEPES buffer (10 mM, pH 7.4)	Live cells	[75]
Zr_6_O_4_(OH)_4_((NDC-(NO_2_)_2_)_6_	390/474	0.1–0.7 mM	20 μM	50 min	HEPES buffer (10 mM, pH 7.4)	Live cells	[76]
Al(OH)(BDC-N_3_)	315/425	0.2–1.6 μM	90.47 nM	60 s	HEPES buffer (10 mM, pH 7.4)	Live cells	[77]
Al_3_O_4_(OH)_4_(BDC-(NO_2_)_2_)_6_	345/527	0.1–0.6 mM	14.14 μM	40 min	HEPES buffer (10 mM, pH 7.4)	Live cells	[78]
Al_3_(O)(OH)(BDC-N_3_)_3_	343/460, 565	0.1–120 μM	100 nM	--	Hank’s balanced salt solution	Cell sample	[79]
Al(OH)(BDC-NO_2_)/poly(vinylidene fluoride)	396/466	0–0.1 mM	92.31 nM	--	PVDF membrane	Lake water	[80]
Cu(TCPP)[AlOH]_2_	419/602, 650	0–10 μM	16 nM	instant	BBS buffer (20 mM, pH 7.4)	Live cells	[81]
CuO@TO@UiO-66	510/~560	0–100 μM	0.51 μM	instant	Tris-HCl butter (20 mM, pH 7.4)	Live cells	[82]
Eu^3+^/Cu^2+^@UiO-66-(COOH)_2_	305/393, 615	0–625 μM	5.45 μM	30 s	HEPES buffer (10 mM, pH 7.4)	--	[83]
Eu^3+^/Ag^+^@UiO-66-(COOH)_2_	305/615	0–2.5 mM	23.53 μM	30 s	HEPES buffer (10 mM, pH 7.4)	Serum	[84]
Tb^3+^@[Cu(CPOC)_2_]	280/390, 544	0–1.6 mM	13.25 μM	2 min	HEPES buffer (10 mM, pH 7.4)	--	[85]
UiO-66-CH = CH_2_ Zr_6_O_4_(OH)_4_(BDC-CH = CH_2_)_6_	328/382	0–0.05 mM	6.46 μM	10 s	HEPES buffer (10 mM, pH 7.4)	Live cells	[86]
Fe^III^-MIL-88-NH_2_	333/~440	60–100 μM	10 μM	5 min	Aqueous solution	--	[87]

^1^ LOD means limit of detection. ^2^ RT means reaction time.

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
