# Peer review of "Recent Progress in Metal–Organic Framework (MOF) Based Luminescent Chemodosimeters"

_nanomaterials, 2019, doi:10.3390/nano9070974_

Reviewer 1 Report

The manuscript by Hao et al. entitled "Recent Progress in Metal–Organic Framework (MOF) Based Luminescent Chemodosimeter" is a review-type article.
This review summarizes recent progress and shows many recent examples of application of porous coordination polymers as chemosensors for several analytes including H2S, F-, HClO, biothiols, redox-active biomolecules, Hg2+, CN-.
Some suggestions:
1. More uniform presentation of MOF examples should be used. Currently different ways of representation of discussed systems are used.
2. The authors should explain more deeply their selection of analytes - why these were chosen? Because of their toxicity or biological role?
3. In section 3.2 the authors introduce AA abbreviation without explanation. Please show the meaning of it.
The presented manuscript shows several recent examples and cites 166 references. The presented material is well-organized and divided into subgroups. This referee believes that the presented manuscript could be published in Nanomaterials after a minor revision.

Author Response

Thank you very much for your consideration on our manuscript! We really appreciate the time and effort that the reviewers have expended on our behalf. Their constructive suggestions and comments have undoubtedly resulted in an improved manuscript. We have revised the entire manuscript based on the Editor and Reviewers’ comments. Below are our point-by-point responses to the questions/suggestions raised by the reviewers.

Reviewers’ comments

Reviewer 1

The manuscript by Hao et al. entitled "Recent Progress in Metal–Organic Framework (MOF) Based Luminescent Chemodosimeter" is a review-type article. This review summarizes recent progress and shows many recent examples of application of porous coordination polymers as chemosensors for several analytes including H2S, F-, HClO, biothiols, redox-active biomolecules, Hg2+, CN-. Some suggestions:

Comment 1: More uniform presentation of MOF examples should be used. Currently different ways of representation of discussed systems are used.

Response: Thanks very much for the referee’s comments and constructive suggestions. Accordingly, we have unified the presentation of MOFs in the manuscript. All types of MOFs were expressed as their molecular formula for the first appearance and were also corresponded with their abbreviated names (unless that the cited article has not provided the specific structure or the molecular formula). Then the same types of MOFs were expressed as their abbreviated names, such as UiO-66.

Comment 2: The authors should explain more deeply their selection of analytes - why these were chosen? Because of their toxicity or biological role?

Response: This review article aims to summarize the recent progress in the development of MOF-based chemodosimeters. Though this kind of luminescent MOF sensors are expected to be highly selective as they exploited the specific reactivities of certain target analytes. Up to now, only dozens of papers have been reported on MOF-based chemodosimeters, which involving the detection of H2S, HClO, biothiols, redox-active biomolecules, fluoride ions, Hg2+, CN-. The present manuscript summarized all these studies and the importance of each target analyte was also disscused in the main text.

Comment 3: In section 3.2 the authors introduce AA abbreviation without explanation. Please show the meaning of it. The presented manuscript shows several recent examples and cites 166 references. The presented material is well-organized and divided into subgroups. This referee believes that the presented manuscript could be published in Nanomaterials after a minor revisions.

Response: Thanks very much for the referee's positive and encouraging comment. Accordingly, we have revised this section title to "3.2 MOF-based Chemodosimeters for Ascorbic Acid" and presented the full name before the abbreviation, "...ascorbic acid (AA)".

Your help will be really appreciated.

Thank you very much in advance for your careful consideration!

Best wishes,

Yanli Zhou

Henan Key Laboratory of Biomolecular Recognition and Sensing, College of Chemistry and Chemical Engineering, Shangqiu Normal University, Shangqiu 476000, China

Reviewer 2 Report

- My main concern is that some parts of the text sound too similar to cited and/or existing references

- Written well, but the numerous errors distract from the quality of the paper; need to proofread from the start; here are some examples:

    - title should likely be plural or Chemodosimetry

    - Figure 5 error - (4) image of A549 cells incubated 174 with 500 μM SNP

    - Missing info in Figure 7

    - UiO, O should be capitalized in paragraph at line 328

- In the Intro, the authors list cerium outside the lanthanide series?

- Otherwise, overall, the manuscript looks to cover a variety of unique areas of sensing and is a nice, broad review.

Author Response

Thank you very much for your consideration on our manuscript! We really appreciate the time and effort that the reviewers have expended on our behalf. Their constructive suggestions and comments have undoubtedly resulted in an improved manuscript. We have revised the entire manuscript based on the Editor and Reviewers’ comments. Below are our point-by-point responses to the questions/suggestions raised by the reviewers.

Reviewers’ comments

Reviewer 2

Comment 1: My main concern is that some parts of the text sound too similar to cited and/or existing references.

Response: Thanks very much for the referee's kind reminding. In this review article, we focused on systematically summarizing the recently reported MOF-based chemodosimeters, including the preparation processes, sensing mechanisms, and the analytical performances for these MOF chemodosimeters. When writing the paper, we also have tried our best to present these examples in our own words. The repetition rate of the main text for this manuscript were also checked using anti-plagiarism iThenticate system, which shows a similarity index of 13% (Shown in appendix PDF file).

Comment 2: Written well, but the numerous errors distract from the quality of the paper; need to proofread from the start; here are some examples:

- title should likely be plural or Chemodosimetry

- Figure 5 error - (4) image of A549 cells incubated 174 with 500 μM SNP

- Missing info in Figure 7

- UiO, O should be capitalized in paragraph at line 328

- In the Intro, the authors list cerium outside the lanthanide series?

- Otherwise, overall, the manuscript looks to cover a variety of unique areas of sensing and is a nice, broad review.

Response: Thanks for the referee's careful checking and constructive suggestions. We have revised the manuscript according to these suggestions.

- In title, "Chemodosimeter" has been revised to "Chemodosimeters".

- In Figure 5, "(4) image of A549 cells incubated with 500 μM SNP..." has been corrected to "(5) image of A549 cells incubated with 500 μM SNP..."

- In Figure 7, the figure caption has been revised to "...(5) topological representation of the 3D structure..."

- Uio-Eu-L1 has been revised to UiO-Eu-L1

- ''MOFs with metal center of lanthanoid or cerium'' has been revised to ''MOFs with metal center of lanthanoid''

Your help will be really appreciated.

Thank you very much in advance for your careful consideration!

Best wishes,

Yanli Zhou

Henan Key Laboratory of Biomolecular Recognition and Sensing, College of Chemistry and Chemical Engineering, Shangqiu Normal University, Shangqiu 476000, China

Reviewer 3 Report

The presented review covers the rapidly growing field of MOFs used as sensors based on irreversible reactions. The number of research papers published in this area is growing fast and therefore a review with such a focus will be appreciated by people working in or entering the area of research. The review is well organized with number of schematic figures depicting the mechanism of action for each described system.

I have only few comments:

1.     The abbreviations AA and HT are not explained in the text or in the list of abbreviations

2.     In line 483 the authors write that “UiO-66@Butyne displayed a identical crystal structure”. This can hardly be true.

3.     MOFs are generally not very stable in aqueous solutions and if luminescent linkers would start leaching from the MOF the sensor would lose its selectivity. Could the authors elaborate also on what requirements MOFs need to fulfil in order to be suitable as chemidosimeter?

Author Response

Thank you very much for your consideration on our manuscript! We really appreciate the time and effort that the reviewers have expended on our behalf. Their constructive suggestions and comments have undoubtedly resulted in an improved manuscript. We have revised the entire manuscript based on the Editor and Reviewers’ comments. Below are our point-by-point responses to the questions/suggestions raised by the reviewers.

Reviewers’ comments

Reviewer 3

Comment 1: The presented review covers the rapidly growing field of MOFs used as sensors based on irreversible reactions. The number of research papers published in this area is growing fast and therefore a review with such a focus will be appreciated by people working in or entering the area of research. The review is well organized with number of schematic figures depicting the mechanism of action for each described system. I have only few comments:

The abbreviations AA and HT are not explained in the text or in the list of abbreviations

Response: Thanks very much for the referee's comments and suggestions. Accordingly, we have presented the full names before their abbreviations. "...for sensing ascorbic acid (AA)..." and "...for sensing 5-hydroxytryptamine (HT) and 5-hydroxyindole-3-acetic acid (HIAA)..."

Comment 2: In line 483 the authors write that “UiO-66@Butyne displayed a identical crystal structure”. This can hardly be true.

Response: We have revised this expression to "The prepared UiO-66@Butyne retained the crystalline structure of MOF UiO-66...".

Comment 3: MOFs are generally not very stable in aqueous solutions and if luminescent linkers would start leaching from the MOF the sensor would lose its selectivity. Could the authors elaborate also on what requirements MOFs need to fulfil in order to be suitable as chemidosimeter?

Response: Thanks very much for the referee's valuable comments and suggestions. Exactly as stated by the referee, many MOFs are unstable in aqueous media, as a consequence only several types of MOFs have been used for constructing MOF-based chemodosimeters, such as UiO-66, MIL-101, MIL-53, etc. According to referee's constructed suggestion, we have stressed the typical features of MOFs which are required for the development of MOF-based chemodosimeters. In the Conclusion section, "And the MOF scaffolds exploited for constructing luminescent chemodosimeters are normally required to possess several features, including high stability and dispersibility in aqueous (or aqueous-containing) media, feasibility of installing the recognition moiety for certain analyte, environmental friendliness and/or good biocompatibility for biosensing and bioimaging applications. "

Your help will be really appreciated.

Thank you very much in advance for your careful consideration!

Best wishes,

Yanli Zhou

Henan Key Laboratory of Biomolecular Recognition and Sensing, College of Chemistry and Chemical Engineering, Shangqiu Normal University, Shangqiu 476000, China
